# What can be learnt from a qualitative evaluation of implementing a rapid sexual health testing, diagnosis and treatment service?

Ava Lorenc [1,2] Emer Brangan [3] Joanna M Kesten [1,2,4] Paddy J Horner,[4,5] Michael Clarke,[5] Megan Crofts,[5] Jonathan Steer,[6] Jonathan Turner,[6] Peter Muir,[6] Jeremy Horwood[1,2,4]

For numbered affiliations see end of article.

**Correspondence to**
Dr Jeremy Horwood;
j.horwood@bristol.ac.uk

## ABSTRACT

**Objectives** To investigate experiences of implementing a new rapid sexual health testing, diagnosis and treatment service.

**Design** A theory-based qualitative evaluation with a focused ethnographic approach using non-participant observations and interviews with patient and clinic staff. Normalisation process theory was used to structure interview questions and thematic analysis.

**Setting** A sexual health centre in Bristol, UK.

**Participants** 26 patients and 21 staff involved in the rapid sexually transmitted infection (STI) service were interviewed. Purposive sampling was aimed for a range of views and experiences and sociodemographics and STI results for patients, job grades and roles for staff. 40 hours of observations were conducted.

**Results** Implementation of the new service required co-ordinated changes in practice across multiple staff teams. Patients also needed to make changes to how they accessed the service. Multiple small 'pilots' of process changes were necessary to find workable options. For example, the service was introduced in phases beginning with male patients. This responsive operating mode created challenges for delivering comprehensive training and communication in advance to all staff. However, staff worked together to adjust and improve the new service, and morale was buoyed through observing positive impacts on patient care. Patients valued faster results and avoiding unnecessary treatment. Patients reported that they were willing to drop-off self-samples and return for a follow-up appointment, enabling infection-specific treatment in accordance with test results, thus improving antimicrobial stewardship.

**Conclusions** The new service was acceptable to staff and patients. Implementation of service changes to improve access and delivery of care in the context of stretched resources can pose challenges for staff at all levels. Early evaluation of pilots of process changes played an important role in the success of the service by rapidly feeding back issues for adjustment. Visibility to staff of positive impacts on patient care is important in maintaining morale.

## INTRODUCTION

Rates of sexually transmitted infections (STIs) continue to increase in England despite

## STRENGTHS AND LIMITATIONS OF THIS STUDY

⇒ The 'trial, assess, adapt' strategy (reflexive process of observation, feedback and resulting action) meant that evaluation and implementation occurred in parallel and allowed researchers to capture the active process.

⇒ The evaluation benefitted the staff, as researchers provided ongoing feedback and suggestions for service improvements and provided a space for reflection.

⇒ A strong and trusting relationship between research and clinic staff arose from researcher flexibility and timely responsiveness and allowed good researcher access to spaces, staff and meetings.

⇒ Frequent, regular and extensive physical presence of the researcher in various clinic settings was crucial as much of the process was not documented.

⇒ The patient sample was limited due to recruitment being cut short by the COVID-19 pandemic lockdown, and we only interviewed men due to the pathway being initially implemented for male patients during the evaluation period.

control efforts, with a 5% increase between 2018 and 2019[1]. *Chlamydia trachomatis* (chlamydia) and *Neisseria gonorrhoeae* (gonorrhoea) are the most common, with 226 411, and 70 982 diagnoses reported in England in 2019, 5% and 26% increase since 2018.[2] The rise in gonorrhoea is particularly concerning as first-line treatment effectiveness, which is threatened by the development of antimicrobial resistance (AMR).[3 4] Most STIs are diagnosed through Specialist Sexual Health Services (SSHS), the provision of which is increasingly challenging as funding (via government public health grant) has been steadily cut since 2015.[5]

Chlamydia and gonorrhoea if left untreated may cause pelvic inflammatory disease (PID) in women, which can result in infertility,

ectopic pregnancy and chronic pelvic pain.[6–8] Infections are often asymptomatic, particularly in women, and when they do cause symptoms and/or signs, these are not pathognomonic.[6 7] Nucleic acid amplification tests (NAATs) provide accurate detection. Early detection and treatment help prevent the spread of STIs and the development of complications. Point-of-care testing (POCT; results within 15–30 mins)[9] and rapid STI services (results on the same day) can potentially improve care and reduce costs, due to reduced time from diagnosis to treatment and number lost to follow-up. This can increase testing uptake, improve partner notification rates and enable better and timelier clinician decisions, improving outcomes such as fewer unnecessary treatments and reduced PID risk.[10–13] Patients prefer rapid STI testing[14–16] and are happy to wait at clinic for results. Rapid testing can reduce anxiety[17 18] and improve patient acceptability of services and uptake of testing.[19–22] HIV POCT is well established and preferred by high-risk men who have sex with men (MSM).[23 24] Although studies suggest a limit of 30 min to wait for results,[25–28] experience from our service indicates that patients would be prepared to wait longer than 20 min for their result.[29]

However, much of the evidence is from modelling and hypothetical views of clinicians and/or patients,[10–12 25–28 30] with little real-life implementation evaluation,[31] and rarely considering the complexity of patients' visits including both asymptomatic and symptomatic patients with multiple needs, for example, female contraception. There is an urgent need to evaluate staff and patient preferences and clinical benefits and cost-effectiveness in practice.

In November 2018, a UK SSHS implemented a first-of-its-kind rapid STI testing, diagnosis and treatment service, using a clinic-based Hologic 'Panther' NAAT diagnostic machine. In 2017, the clinic introduced an online STI and HIV testing postal service for asymptomatic patients.[32] The new rapid service provides chlamydia and gonorrhoea results in 3.5 hours (previously over a week when tested in the microbiology laboratory), to improve patient care while reducing costs (see figure 1 for an overview of the service redesign). This evaluation assessed the best service model and patient and staff acceptability, to refine and improve the service and support implementation in other SSHSs. We report the qualitative evaluation of male patient and staff views and experiences of the implementation of the first phase of this new rapid STI service.

## METHODS
### Design
The evaluation was ethnographic, used observations and interviews[33] and was informed by Normalisation Process Theory (NPT). NPT is a sociological theory that has been widely promoted as a means to understand implementation, embedding and integration of innovation in healthcare settings until they become normalised and routine.[34] This approach focuses on actions people perform to normalise an intervention within the contexts and locations they inhabit.[34] NPT proposes that successful implementation

of an intervention is dependent on participants' ability to fulfil four inter-related criteria, which interact with the wider intervention context[34]: (1) coherence—(sense-making—understanding and opinion of the intervention's purpose); (2) cognitive participation (commitment and engagement with the intervention); (3) collective action (the work that individuals and organisations have to do to make the intervention function); (4) reflexive monitoring (appraisal of the intervention once it is in use). NPT supported real-time feedback to refine and improve the service. The study focused on four timepoints selected pragmatically during 16 months of evaluation: T1 at start of implementation; T2 after 6 months; T3 after 14 months; T4 at 16 months during the COVID-19 pandemic lockdown.

### Setting
A sexual health clinic in Bristol (population 450 000), UK.

### Participants
Due to the new service being initially introduced for the male pathway only, male patients (over 16 years old) and staff at the sexual health clinic were interviewed. Patients were invited to take part, via a clinic survey about PrEP (pre-exposure prophylaxis for HIV)[35] and when physically attending the clinic at T1, T2 and T3. Cross-sectional interviews were conducted with staff at four timepoints at T1, T2, T3 and T4. One staff member was interviewed two times. Purposive sampling[33] attempted to capture maximum variation in views and experiences, and sociodemographics and STI test results for patients, and job grades and roles for staff (administrative staff, consultants, doctors, nurses/nursing assistants, health advisers, Public Health England); responsible for the Panther laboratory and administration. Information sheets were provided to male patients by staff at the clinic or via email from researchers, with patients asked to contact the researcher and ask questions before deciding to take part. Staff were emailed by the researcher about the study.

### Data collection
Following the concept of information power, data collection continued until sufficient data to meet the study objectives had been collected with continuous, pragmatic assessment of information within our sample.[36] Issues informing information power includes the study aim (ie, broader aims require a larger sample), the sample (ie, a smaller sample is needed if participants have rich experiences relevant to the research), use of theory (studies supported by theory require smaller sample sizes), depth and quality of the data (ie, smaller samples are needed with focused and clear data) and the analysis type (larger samples are needed for exploratory analysis).[36]

In the first 6 months of service implementation, observations were conducted by EB and JMK at varying times/days, in reception, laboratory and waiting areas. Non-participant observations focused on day-to-day operations, how clinic staff integrated the new service and any

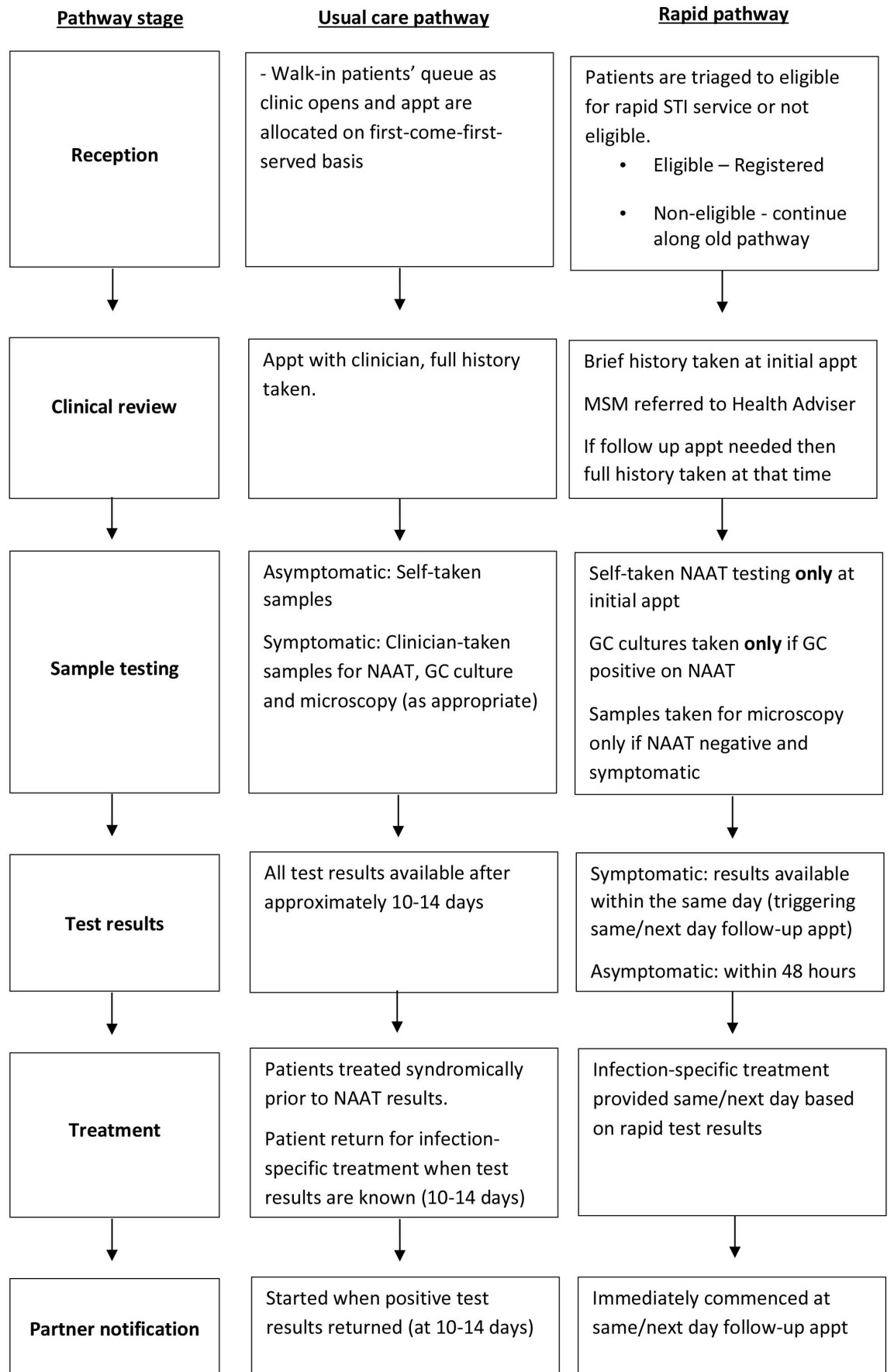

**Figure 1** Overview of rapid pathway service redesign. appt, appointment; GC, gonorrhoea culture; MSM, men who have sex with men; NAAT, nucleic acid amplification test.

factors that promoted or inhibited successful incorporation.[37] Written accounts based on brief field notes taken at the time included observations, conversations with staff and reflection on what had been observed.[38] Observations recorded activities, events, their time and location and described interactions, communication patterns, workflows and tasks in the clinic environment.

Interview topic guides (online supplemental files 1 and 2) informed by NPT explored: views and experiences of the service; impact on workload and clinical practice; information and support needs, sustainability and future implementation of the service. Patient interviews took place throughout the evaluation period and explored their experience and views of the service, including acceptability, barriers and facilitators to uptake. Patients were offered a £10 High Street shopping voucher. Interviews were conducted by experienced qualitative senior research associates AL/JMK/EB, used flexible topic guides and open-ended questioning, were face-to-face (at the clinic or University) or by telephone and lasted around 30 min. Participants were told that the study was evaluating the rapid results service and that interviewers were independent of the service.

## Analysis

Interviews were audio recorded, transcribed verbatim and imported into QSR NVivo (V.10) with transcribed observation fieldnotes. Ongoing and iterative analysis informed further data collection through changes to the topic guide and feedback to healthcare staff to aid the adaptation and refinement of the rapid service. 'Codebook' thematic, inductive analysis by EB/AL identified and analysed patterns and themes salient to interviews and observations.[39] Initial noting of ideas was followed by line-by-line examination and inductive coding. A subset of transcripts and observations were independently double coded by EB/JH and discrepancies were discussed to contribute to the generation and refinement of codes to maximise rigour. Themes were discussed by the multidisciplinary research team to ensure credibility and confirmability. Negative cases and reasons for deviance were explored. The four NPT constructs[34] were used to further develop themes deductively.

## Patient and public involvement

PPI meetings with three people who recently used the clinic informed the study design. These meetings reviewed patient-facing materials and discussed the acceptability of proposed recruitment and data collection.

## RESULTS
### Participants/hours of observation

We conducted 25 observations over approximately 40 hours, 25 staff interviews (24 participants) and 26 patient interviews. Patients were aged 34 years on average (range 19–57 years), most identified as MSM, two had positive STI test results and the index of multiple deprivation scores averaged 5.4 (range 2–10).

### Coherence (sense making)

Staff and patients welcomed rapid testing (table 1). All staff saw it as beneficial and many were excited about doing something new, particularly to improve service access, which was limited by a lack of prebookable appointments, high-observed demand (manifesting in long queues outside the clinic before it opened each morning to access limited capacity walk-in appointments) and staff shortages. Staff welcomed being able to provide treatment based on results and avoiding unnecessary antibiotic prescribing (previously treatment was prescribed presumptively for symptomatic patients due to the week-long wait for test results). However, some staff had concerns around anticipated reduced clinician contact with patients and shorter consultations in the new service. Patients valued a potentially quicker and more convenient service and also reduced anxiety from long waiting times for results. Some patients valued avoiding unnecessary antibiotic treatment for personal and wider societal reasons.

### Cognitive participation (buy-in)

The importance of engaging the whole clinic team in the service redesign was recognised but challenging with a large team and many part-time staff (table 1). Formal engagement was via an implementation team, project meetings and staff training sessions. Engagement of 'on the ground' staff was inadequate, with administrative and nursing staff feeling particularly disengaged and having limited time to prepare for the new service. Staff cited the following issues around engagement: poor communication (due to busy work schedules with limited time for accessing emails), a lack of access to training as many staff were part time and did not work on the day training was delivered or lack of involvement in project meetings or the implementation group, which was initially only senior staff, although the latter improved as the project progressed. Engagement was also limited by a lack of staff-protected project time and a context of burn out from staff pressures (eg, funding cuts, understaffing and high service demand). Implementation work was fitted around existing high workloads and rapid service changes made timely feedback to staff difficult.

### Collective action (putting rapid STI test results service into operation)

The service was implemented for male patients in November 2018 and for all patients in August 2019.

Important in collective action was designing and documenting the new patient pathways, which needed to be clear but flexible to allow staff deviation from protocol to respond to individual patient situations and need (eg, anxiety, medical history, relationship status, availability to attend clinic). Guidelines, Standard Operating Procedures (SOPs) and pathways had to be rewritten as initial

**Table 1** Quotes on coherence and cognitive participation

| Coherence | Cognitive participation |
|---|---|
| **Staff enthusiasm and concerns**<br>It all sounded quite exciting and I was quite, not excited, but it was like, this is really good, the first one in the UK and, you know, this will be excellent, so I was quite open minded about it… lowering the use of antibiotics and fewer invasive procedures for women. (Nurse 17, T3)<br>It's an exciting opportunity, um, but also there's a bit of, um, er, you know, nerves I suppose about how it will actually be, actually run from day one, really, and how it would go (Public Health 14, T2) | **Disengaged staff**<br>It was very much the higher-up staff that kind of organised it all and they're not really the ones that are going to be doing the actual work, so I think it's really important to include clinical staff of all levels, kind of when it's getting near to it and you know, really explain, and have their opinions and thoughts on, you know, how it's gonna work (Nurse 16, T2) |
| **Reduced patient anxiety**<br>Results on the same day would be amazing, yeah, no doubt about that yeah, because there's always quite an—well, for me, it's like an anxious wait otherwise. Yeah, that sense of not knowing and actually "how do I manage my sex life in case anything comes back positive?" Yeah, quick results can definitely make a big difference (Mike, didn't use rapid STI service)<br>The thing that would encourage me to test more regularly, [is] if the whole thing [time spent in clinic] could be done in a shorter time slot for me (Ben, used rapid STI service) | **Barriers to engagement**<br>(Clinic managers need to] canvass people's feelings about it because I don't know that we do that very well. I think we just crack on. We don't say 'how was it for you that first week did you cope? Did you keep your head above water?' (Nurse 10, T1) |
| **Avoiding inappropriate antibiotics prescriptions**<br>With antibiotics I'm really—they really mess with my stomach. I really feel really sick whenever I do take them. So, I'm just pro not taking them for that reason alone. Yeah, like—I try to think about the bigger picture of the world and stuff…but I think about my own stomach more than the wider world. So, yeah, I'd—I'm always pleased to like not have to do any unnecessary drugs (Andy, results by text)<br>I just think I'd hate to kind of like take something that I didn't have and then if I ever got it again it doesn't work—so it's kind of like half society view half personal view (Harry, follow-up appointment) | **Inadequate preparation**<br>We were told on the Wednesday that it was supposed to be starting on the Monday and we were like all a bit shocked thinking 'well hang on a minute what about the training? We'll look really, really stupid in front of patients' (Administrative staff 6, T1)<br>There was always talk about what it [Panther] could do, never talk about how it's going to function. Even at the last minute, the week before it was meant to start [Lead Consultant] came to me and goes, 'This is what I think the pathway is. Can you make notes on it?' Nobody was really clear about what happened. [Project manager] had sent a PowerPoint around and it was embedded in a way that… some staff who aren't maybe familiar with how embedded links and things work(- couldn't open it). When the meeting came around, they said, 'Has everyone read the email about the Pathways?' half of the people went, 'What email?' …, it just wasn't presented to us in a clearest way. (Health Adviser 1, T2) |

implementation issues were resolved. However, detailed SOPs were not always in place prior to implementation of a new modification to a pathway, making it difficult for staff to keep up with current processes. This was due to the repeated and frequent changes to clinic processes/ patient pathways to resolve initial implementation issues, and the lack of protected administrative staff time meant that when patient pathways were revised following staff feedback, these could take over 6 weeks to review and be signed off by the clinical governance group. For example, the triage form asking patients to self-identify at reception whether they were symptomatic, their risk level, and if they had had sex against their will was revised three times during implementation to make it clearer, which pathway should be followed. Observation showed that they annotated a copy of the triage form to remind them of the pathways for different responses. In contrast, patients were happy with communication about the changes made (via the website, staff, consultations, on the triage form).

Although staff accepted the continual adaptations as inevitable due to the novelty of the service, it was difficult. During initial implementation, 'teething issues' were experienced, including administrative staff not knowing which patients were eligible for the service, dealing with the high volume of patients when the doors first open, and the best way to triage patients. Staff worked together to adjust and improve the new service, identifying problems and opportunities and innovating in their own practice, overseen and supported by the implementation groups, and morale was buoyed by the positive impact on patient care and the positive feedback from the research team.

The evaluation process played an important role in the success of the service by rapidly feeding back issues for resolution. The evaluation process aided communication, and researchers were able to suggest solutions to problems based on the non-participant observation. For example, researcher (EB) codeveloped with the clinical team a

**Table 2** Changes along the (walk-in) male patient pathway

|  | Before rapid STI service | Rapid STI service | |
| --- | --- | --- | --- |
| Registering at reception | All walk-in patients allocated an appointment in order of queue—may have to come back later that day | Receptionist triages each patient to appropriate pathway referring to guide/pathway | |
|  |  | Patients not eligible for rapid STI service are given an appointment for later that day and continue on old pathway | Rapid STI service-eligible patients wait after registering to be called up |
|  | Triage form used to register patients | Triage form amended to be gender neutral and to make categories clearer | |
| Seeing a clinician/ health adviser | At first appointment | First appointment very brief—reduced history taking. Unless uncomplicated vaginal discharge | |
|  | MSM and all patients needing partner notification/risk reduction/safer sex advice see health adviser | High-risk patients (MSM or new to service) see health adviser at initial appointment | |
|  |  | Symptomatic men see doctor/trained nurse only at follow-up not drop-off, usually on same day | |
| Providing samples | Urine and swabs: taken at time of consultation. Self-taken if asymptomatic. If symptomatic: clinician taken swab for microscopy (to detect NGU* and gonorrhoea) and gonorrhoea culture; NAAT self-taken | Self-sample drop-off—urine and swabs NAAT self-sampled in toilets, putting samples through a hatch to the laboratory. Instructions are on posters in the toilet and from nursing staff/NAs. Gonorrhoea culture taken by clinician on return only if NAAT positive. Swab for microscopy to detect NGU if NAAT-negative (see text above) | |
|  | Blood samples taken by doctor/nurse/nursing assistant (NA) | Blood samples taken by doctor/nurse/NA | |
| Tests results | All STI test results in 2–3 weeks | Chlamydia and gonorrhoea processed on Panther—results within 48 hours. Others still 2–3 weeks | |
|  | If negative, text sent | Results by text if asymptomatic and negative | |
|  | If positive, HA phones patient to discuss result and arrange treatment (unless already received presumptive treatment) | Results given at follow-up appointment with clinician if symptomatic or asymptomatic positive | |
| Treatment | Treat presumptively | Wait for results before treating | |
|  | Treat at first (only) appointment | Treat at follow-up appointment | |
| Notified partners | Treated immediately | Only treat on positive results (if sexual contact was >2 weeks; otherwise treat immediately) | |

*NGU (Non-Gonococcal Urethritis) is an infection of the urethra caused by pathogens (germs) other than gonorrhea.
MSM, men who have sex with men; NA, Nursing Assistant.; NAAT, nucleic acid amplification test; STI, sexually transmitted infection.

laminated card for patients explaining the new service in response to the researcher observation that patients were given variable information by reception staff. Some of the changes to the patient pathway (table 2 and box 1) caused challenges. The responsive model meant comprehensive preparatory training and communication to all staff was challenging, and multiple methods of communication were essential. Many staff found changing ingrained behaviours difficult, particularly reducing the content and duration of consultations when they had been taught to maximise patient contact. The shorter initial appointments, with reduced medical record completion and fewer physical examinations, were a 'huge change' and source of concern and anxiety for clinicians both before and during the changes, due to perceived loss of opportunities for patient discussions about domestic violence, female genital mutilation, alcohol use, contraception, etc, which are seen as essential for a 'holistic', 'integrated'

'level 3 service'. This did improve with practice, and patients with particularly concerning issues were referred for a health adviser consultation, which was longer under the new service. Self-sampling drop-off also meant reduced clinical contact, particularly for asymptomatic, low-risk men with negative test results (health advisers only see high-risk/new patients with MSM at the first visit). The walk-in clinic was, therefore, more demanding, as the case mix changed, seeing more symptomatic patients with complex presentations. Although reduced clinical contact with asymptomatic patients was a planned cost-saving benefit, it meant nursing assistants (running the sample drop-off sessions) collected mandatory GUMCAD STI Surveillance System Dataset (GUMCAD) data[40] and answered patient clinical queries, which they were not qualified/paid/willing to do.

In the new service, chlamydia and gonorrhoea treatments were to be given based on results, not presumptively

## Box 1 Collective action quotes

**Designing and documenting new processes**
We need to be really clear about what we're doing when they drop-off(patients drop-off samples), what we're doing when they come back, what are we going to do about contraception, which questions are okay to leave out of the proforma… for consultants because we have a lot of experience and because we're used to making decisions then I think we can [unclear] a bit and we can be flexible and can you know think about the individual patient. But for nurses who work a lot to PGD's [guidelines] and like to have clear guidance. And some of the juniors as well, who will be quite new—you know they've just been changed. Because it's going to be bewildering and chaotic you know it's doesn't feel good when there's chaos on the shop floor (doctor 3, T1).

**Flexibility in pathways**
It's a case by case situation and it does help to have helpful medical staff that have been willing to make an exception (administrative staff 18, T2).
I think the health advisers also are more able to …know the guidelines but in some situations know that you have to approach things differently, … for me personally if I was seeing someone and they kind of said 'actually I have got this dis(ease)'—you know, real clear symptoms, you know, 'and I'm really fed up with it', I'd be more inclined to say 'okay then let's get you treated…I think you can have your general thing of saying to someone 'look you know come back in the afternoon' but if you've got someone who's kind of 'actually no but I've had these symptoms for two days I've really had enough of it' (health adviser 13, implementation group member, T2).

**Guidelines**
It's a work in progress but the problem is as the pathway evolves then the guideline will change again…because this is so rapidly moving actually, I don't think I really want to do a guideline. So it's kind of hard to have a guideline anyway but we need some kind of guidance (doctor 3, T1).

**Teething issues**
It was chaos, the first few weeks were chaos. Reception didn't know what they were doing… there was hundreds of patients around the reception, we didn't know what we were doing, so yeah, it was chaos, but it has slowly got better (nurse 17, T2).
The waiting area fills up and people are filling out the Panther triage forms on windowsills. After a while [clinic coordinator] tells receptionists on Panther desk that he had given out 16 forms. Once they get to 10 people booked for Panther returner pathway they need to go check with the laboratory regarding further capacity (observation notes, reception area).
Two reception staff were unsure whether one person should be panther/same day or walk-in due to the information provided on the form. Staff consulted with person entering data on computer. They checked whether person was returning for results/treatment. They explained to the patient that a new system is in place, so they want to make sure they do the best for him (observation notes, reception area).

**Understaffing**
It's been very stressful for staff and I think it has been an enormous amount of work for the implementation group, that I think in the private industry you'd be given huge amounts of time, whereas we virtually squeezed it in amongst everything else we've done, but that's just the NHS (doctor 11, T2).
Administrative staff/reception team has three staff vacancies, and today there are two members of clinic staff off sick—one clinician, and one

Administrative staff (clinic coordinator). The clinician would have been doing sample drop-off, and walk-in, so have had to reduce slots for both until they get confirmation of clinical capacity from clinicians when they arrive (observation notes, reception area).

**Changing ingrained behaviours**
It's been quite hard on staff and obviously there's a lot of—you know, if you've been doing something the same way for 10/20/maybe 30 years, that's quite a massive change for people (nurse 21, T3).

**Changes to clinician contact**
We've actually ended up seeing a lot more complicated or complex patients, at least that's how it feels. The easy patients get siphoned off quite quickly and that means that more patients [can be seen), especially the complex patients, which the nurses are less able to deal with and require a lot more consultant supervision. I think there has been a general feeling in the department that the consultant cover job is busier than it ever was before (Doctor 11, T2).

**Challenges of changes at reception**
(Receptionists) were worried that they were looking like they didn't know what they were doing, because it was new and they weren't quite sure. So I think it took a, it was a lot to ask for them all really because it was a big change, but it is just that keep reminding everybody that actually, in the long term, you will get it, and it's much better for other patients once it's in place (administrative staff 12, implementation group member, T2).
Reception staff on male desk refresh the panther decision pathway together using A4 sheet. Female staff member commented that she always has to double check the process. Reception staff discussed male staff member's confusion about eligibility for Panther (observation notes, reception area).

**Concern about shorter consultations**
It was a huge change, because, we, it is quite a detailed consultation. We have been told time and time again that 'oh you need to ask patients about domestic violence, ask the women about female genital mutilation, you need to do this'. Then all of a sudden, they are saying, 'no, don't ask any of these things', it's like aargh! (nurse 17, T2).
It does sometimes feel if I'm absolutely honest a little bit less than a level three service, you know, people are just coming in and dropping off a sample. I know that's possibly better use of our time, but it seems a little bit spurious to call it level three (nurse 10, T1).

**Perceived patient views on waiting for treatment**
I think the major anxiety that patients have is around not being treated immediately and not being treated necessarily as a contact of infection and anxiety around that. I often find that with a bit of educating that that is overcome and my major impression is that patients really appreciate it (doctor 11, T2).
I haven't had anybody who's been absolutely, you know, anti about it but there have been a couple of people who I've thought 'I'm going to treat you mate, I'm not going to wait on results' do you know what I mean?… they're anxious, they've maybe got another partner, a regular partner, who they don't want to infect, which, you know, I can see the reasoning behind that. But I think, you know, once that kind of idea has got out amongst our regular clientele I think it will be a lot easier (nurse 10, T1).
Four young men approach the door together. [Name] lets them know that he has just 2 forms/slots left at present, and suggests that he gives these to them on the basis of who came through the door first—these two seem pleased, and head in with their forms. He asks the other two

**Box 1    Continued**

to wait here for a minute and they seem OK with this. He tells them there is a new service which means they can give people results/treatment faster, and this is why things are different. He asks them to give him a yes/no answer as to whether they have any symptoms. When they say no he says that it is probably not worth them waiting as they are unlikely to be seen today, and that they could come back another morning for when the doors first open. They seem to find this acceptable (observation notes, front door).

Benefit of evaluation process
Staff member commented that it was helpful to have an outside voice (research team) feeding back, because sometimes when you are within the structure you can be shouting stuff and nobody hears you (observation notes).

unless sexual contact with a case was within the 2-week window period and patients requested treatment.[6 7] Men with symptoms of urethritis were first tested for chlamydia/gonorrhoea and booked to return more than 4 hours later. If NAAT-positive, they were treated according to British Association for Sexual Health and HIV (BASHH) chlamydia and gonorrhoea guidelines[6 7] and if negative tested for urethritis and managed according to BASHH guidelines[41] (with reassurance, including a leaflet, if negative) and told to reattend for an early morning smear if their symptoms did not resolve. Some patients, particularly regular clinic attendees, were initially not keen on this longer wait for treatment, although this did improve. A minority of clinicians deviated from protocol and treated presumptively, especially for patients who were particularly anxious. Staff reported mixed patient understanding of only treating when results were available, with detailed explanations needed, but patients were amenable once they understood.

### Reflexive monitoring (appraisal of STI test result service into operation)
#### Contextual factors
In addition to the contextual factors described above of inadequate service funding, understaffing and, ongoing communication problems, increased use of postal testing (meaning less complex patients used postal testing and more complex patients used the walk-in clinic) and increasing use of PrEP increasing service demand. Increasing societal awareness of gender issues also influenced the service experience, with triage forms issued to male patients on arrival creating tensions around sensitively managing patients who did not identify with their sex assigned at birth (including trans and non-binary patients). This process was amended following feedback from the research team.

#### Success
Overall, the new service was seen as successful as it was implemented and running fairly smoothly after initial problems (box 2). Although the process was challenging,

**Box 2    Reflexive monitoring quotes**

**Success**
When you speak at the national [sexual health] meetings, people, it's a bit of a no-brainer, what we're supposed to do, and people are amazed that we've been able to introduce it [rapid STI service] cost-neutrally. Because when you look at the point-of-care systems which other people are researching, it's… more expensive, so we've adopted an innovative approach (doctor 19, T3).

**Quality of care**
The person I saw was really brilliant, like, yeah. I felt really comfortable… I really felt like I could ask anything I want and felt sort of safe (Andy, results by text1).

**Benefits**
It was the immediacy and the kind of reassurance that … if something was positive that you would be able to treat it straightaway (harry, follow-up appointment).
They're [patients] very happy. I mean, who wouldn't be? You find out the same day that you have got chlamydia and you can start your treatment. I mean that is brilliant (nurse 17, T3).
Dr 11: One of the advantages that we hoped would come out of introducing Panther would be that we would attract more high-risk people, because it would be seen as an attractive place to come and test, and also that it would free up staff time so that we could spend more time with risk reduction etc.
Interviewer: Do you think that is happening? Or is it not there yet?
Dr 11: I think it has started to happen, I don't think that's only down to Panther, I think that's down to some other stuff like PrEP and things like that as well. I feel like the cohort of patients that we see is increasingly complex (doctor 11, T2).
I think the most positive things are seeing your symptomatic patients with knowing what is going on with them. You know what infection they have, you know what treatment they require or if they don't have anything you can then take the time to discuss that (doctor 11, T2).
I think it's [rapid service implementation] made the staff more able to deal with change [to telephone clinics], because they had undergone experience of change with Panther pathways over the past 12–18 months…. Yes, it probably made it smoother and more efficient (doctor 22,T4).

**Suggested improvements**
The main issues that have arisen have been when the [Panther] machine fails and that can be pretty catastrophic (laughs), just because you have booked slots and patients come back and you don't, you can't even tell them whether they have chlamydia or gonorrhoea and they've, kind of, come with that expectation (doctor 20, T3).
Interviewer: If you had an imaginary clinic who were going to set out on this path, what would your advice be to them overall? With the knowledge that you've now gathered from your experience, is there a way that you could help them?
Dr 11: I think preparation, preparing all of the documents that support your staff on a day-to-day basis, clarifying your communication pathways, giving your lead clinicians adequate time in their work plans to do all of that. I would definitely support it, I think it's definitely been a major benefit (doctor 11, T2).
It has been a big change for all staff working, and it's difficult to know whether there was any way of realising some of the things we hadn't realised. I don't think we could have done. I think they were literally just things of implementation that have caused some additional tweaks required - and that in itself has been stressful because it's been the

**Box 2    Continued**

realisation of what are we doing in this scenario and not being quite prepared for it (administrative staff 12, T2).

The thing that I found most interesting is the communication difficulties in amongst the staff and how difficult that has been, having an implementation group that I think represents most of the groups that it's impacted upon and the difficulty that the messages just have not got to the clinic floor, and that's an on-going issue (doctor 11, T2).

implementation was an achievement, given the constraints on resources and staffing and lack of additional funding highlighted above. Staff were credited with being adaptable, highly motivated, hardworking and mutually supportive. Staff's job satisfaction and morale were boosted from doing something new and exciting and they felt proud about achieving implementation, which contributed to enhanced teamwork and coherence. Better job satisfaction was mainly due to improvements to consultations with patients, including consultants seeing more complex patients. These boosts gave the team confidence that they could make further service improvements, demonstrated by the rapid changes made during the first wave of the COVID-19 pandemic during which staff reported being more 'change-ready'.

Although staff were initially concerned that the changes would jeopardise the quality of care, this does not appear to have been realised and patients felt very positive about staff and the ability to raise concerns and discuss issues. Staff perceived that the service was able to see more patients, and that clinicians and health advisers could spend more time and better engage with complex and higher risk patients due to more efficient processing of patients attending for routine testing. Self-testing and fewer physical examinations involving invasive sampling (urethral swab) were generally preferred by patients. Decreased time to diagnosis and treatment meant less patient anxiety while waiting for results and most patients were happy to wait up to 48 hours for treatment. Indeed, patients rated the quality of the new service highly, with some patients specifically requesting it. Staff, and some patients, were pleased to be able to treat with results, which promoted informed discussions and reduced antibiotic use, secondary complications and onward transmission.

### Suggested improvements

For many staff, the most important improvements to the implementation were preparation of documentation of new processes and pathways as soon as possible and engaging and supportive communication from senior staff with all staff but particularly nursing and reception teams to improve process design iterations. This communication should use a variety of methods (especially face-to-face) including written, training sessions, on-the-job support, informal and nominated individuals for support. Bringing teams together for training was recommended to facilitate information exchange and understanding.

It was recommended that, if possible, staff needed to be better prepared for behaviour change and multiple continual adaptations. Staff also need protected time for the project, and the impact on staff roles and workloads needs to be better considered. Small-scale pilots of the new service with patients, to test and refine draft processes to reduce staff stress and confusion, were proposed. Other areas for improvement were consistency in the rapidity of results and contingency planning for malfunctions (sometimes, results were not available on time due to Panther machine breakdowns); more and earlier information for patients, especially on the process and timings (waiting times, results notification, etc). Finally, the use of phone/video clinics, which were implemented during physical distancing requirements of COVID-19, may have benefits elsewhere.

The online supplemental file 3 summarises service considerations for implementing a rapid STI service and relevant teams/job roles.

## DISCUSSION
### Principal findings

The first UK rapid NAAT testing integrated SSHS for chlamydia and gonorrhoea was successfully implemented despite funding and staff shortages. Inevitable initial challenges were resolved and, overall, it was well received. Staff were enthusiastic about it and understood the benefits, although some were concerned about reduced patient contact. The use of NPT allowed for examination of issues with both the design of the rapid service and its implementation. Cognitive participation difficulties included engaging all staff and changing ingrained behaviours (resulting from extensive training and audit), especially for administrative and nursing staff, although staff did support each other and work together. Some patients had concerns about waiting for treatment, but most accepted sample drop-off and returning for a follow-up appointment. Reflexive monitoring revealed perceived benefits, including reduced patient anxiety, seeing more patients and boosting staff job satisfaction. Infection-specific treatment based on test results was crucial, enabling informed consultations and improving antimicrobial stewardship. Suggestions for this and other future services included: documenting new pathways and processes early and comprehensively disseminating to staff; involving all staff in planning, design and implementation; protecting staff time for meetings and actions; considering pilots with a small group of staff/patients before sharing more widely or writing guidelines; cross-discipline training; varied methods of, sensitive and supportive communication; considering staff role impact and ensuring staffing to cover changes.

### Relation to other studies

Evaluating the real-life implementation of a novel rapid results service confirms previous hypothetical/simulated studies where patients were happy with the service and

willing to wait for results before treatment,[14–16 25] whereas previous research has found that the patients found the hypothetical scenario of waiting up to 40 min for test results acceptable,[25 26] our findings demonstrate that patients were happy to wait up to 48 hours for treatment based on results. Willingness to wait has been found to be dependent on self-assessed infection risk and anxiety about their infection status.[25] Our findings demonstrate that the rapid service can lead to less patient anxiety due to shorter time waiting for results and, therefore, should target patients concerned they are infected. Although asymptomatic patients are encouraged to use online postal services, some patients may wish to attend in-person clinics.[13 42] The previously anticipated[12 18] benefits of treating with results and improving antimicrobial stewardship are highly valued by staff and patients in our evaluation. Modelling studies have demonstrated that rapid testing can enable faster treatment, reduces infectious periods and leads to fewer transmissions, partner attendances and clinic costs.[43 44] Rapid diagnostics and treatment can increase the proportion of individuals receiving timely treatment and decrease community prevalence of STIs[45 46] and recently has been seen as a key factor contributing to reducing new HIV infections in London and ensuring those with HV receive fast and optimal care.[47] Our findings also confirm reductions in patient anxiety[12 17 18] and improved testing uptake[19–21] are likely, as well as freeing up clinician time, greater clinician confidence, and efficiencies allowing capacity to be used elsewhere.[12]

The challenges of communicating with and engaging all staff, especially those 'on the ground', and the need for dedicated time for training and implementation[48] are key in healthcare quality improvement.[48] Teething issues experienced in this service—documentation of new pathways, impact on staff roles—and the challenges of changing ingrained behaviour—are common in implementation of a major service change and emphasise the importance of staff training and communication of the reason and implications for change.[49]

Our findings demonstrate that successfully implementing a beneficial service change can boost staff job satisfaction and morale. Previous research has found improvements in staff satisfaction following successful sexual healthcare innovation.[49] This finding suggests that the implementation realised benefits for staff—previously highlighted as influencing acceptance of change in NHS service improvement programmes[50]—and aligned with professionals values and intrinsic motivation to provide quality and effective care.[48]

### Implications

This study shows that a rapid NAAT-testing integrated SSHS for chlamydia and gonorrhoea can be implemented in a constrained NHS system and is acceptable to patients, with benefits for staff, patients and public health, including reduced patient anxiety. The perceived efficiency (to be clarified in a separate quantitative evaluation) is crucial given the financial and staffing pressures on UK sexual health services.[51] Similarly, the pride of staff in their service and enhanced staff satisfaction are important in boosting staff morale and are likely to further enhance the provision of high-quality patient care when such a service is introduced.

AMR is a major concern for gonorrhoea, and a priority worldwide[52] and in England.[53] Rapid STI services could play a vital role in reducing unnecessary antibiotic prescribing by providing test results during/soon after consultations, allowing informed clinician choices. When the technology becomes available, the addition of POCTs to detect ciprofloxacin-sensitive gonorrhoea will dramatically reduce reliance on ceftriaxone and selection pressure for AMR.[54 55]

Our implementation recommendations for future services echo those from the Health Foundation, such as sensitive leadership oriented towards inclusion, agreeing to roles and responsibilities at the outset and 'bringing everyone along with you'[48] as well as early documentation, piloting pathways, varying communication methods and adequate staffing. The willingness of symptomatic male patients to wait for treatment can inform development of new care pathways using POCTs,[12 56] although results are limited to a single service and male patients.

### Strengths and limitations

Project strengths include: integration of findings from multiple qualitative methods generating rich insights, a multidisciplinary team including clinical academics; a strong trusting relationship between research team and clinical staff due to existing relationships and research team flexibility and responsiveness; regular feedback from researchers to clinicians using a 'trial, assess, adapt' strategy. EB and JMK came to the observations as experienced researchers and with good knowledge of the plans for the service changes and reasons for them. The researchers were surprised at how quickly it was possible to provide information and feedback to the implementation team, which they clearly valued highly and rapidly implemented changes based on it. The researchers could move freely between different physical areas of the clinic and stages of the process in a way which clinic staff were not free to do, which provided early insights. Due to the study design and relationships, these insights could be discussed promptly with relevant staff—and so sense checked, and action taken in response if appropriate (changes to clinic processes; further data collection, etc). The rapid, supportive, evidence-based feedback which the researchers could provide or seemed to quickly build the confidence of the key implementation staff in the research process. The researchers appeared to be quickly accepted as trusted team members, with the capacity to help with the work at hand (rather than creating 'research burden').

Limitations include an all-male patient sample as the service was initially only for males, and when implemented for females, few were eligible and evaluation

was hampered by the COVID-19 pandemic. We aimed to include patients with positive STI results but most (although symptomatic) were negative, limiting evaluation of follow-up appointments. COVID-19 meant fewer final batch interviews. As the rapid STI result technology develops, continued implementation evaluation is important,[56] to capture the wide-ranging impact on services, staff and patients. Evaluation for female patients is needed, given the challenges around contraception and STIs/symptoms.

## CONCLUSION

As the first UK SSHS to implement rapid NAAT testing for chlamydia and gonorrhoea within an integrated service, this project faced the challenge of innovating to save time/money and improve patient experience in a constrained environment, particularly lack of funding and understaffing. Inevitable challenges—mainly related to the impact on patient pathways—were resolved and, overall, it was a success. Perceived benefits included reduced patient anxiety, seeing more patients, treating with results, reduced antibiotics use and boosting staff job satisfaction. Learning for other services considering implementing something similar includes more inclusive staff engagement, sensitive communication, better documentation of changes, dealing with constant adaptations, and consideration of the impact on staff and their roles.

**Author affiliations**
[1]Population Health Sciences, Bristol Medical School, University of Bristol, Bristol, UK
[2]National Institute for Health Research, Applied Research Collaboration West (NIHR ARC West), University of Bristol, Bristol, UK
[3]Department of Nursing and Midwifery, University of the West of England, Bristol, UK
[4]NIHR Health Protection Research Unit in Behavioural Science and Evaluation, Population Health Sciences, Bristol Medical School, University of Bristol, Bristol, UK
[5]Unity Sexual Health, University Hospitals Bristol and Weston NHS Foundation Trust, Bristol, UK
[6]South West Regional Laboratory, National Infection Service, Public Health England, Bristol, UK

**Acknowledgements** We would like to thank all study participants.

**Contributors** JH, PJH, EB, JMK were responsible for the study design. JH and EB were responsible for study management and coordination. EB, AL, JMK and JH led data collection and analysis. MCl and MCr co-led the development and implementation of the new service model. PM, JS and JT supported implementation and accreditation of Point of Care testing. MCl, MCr, JS, JT, PM supported the study design and interpretation of interview findings. All authors read, commented on and approved the final manuscript. The guarantor is JH.

**Funding** Funded by the National Institute for Health Research (NIHR) Applied Research Collaboration (ARC) West at University Hospitals Bristol and Weston NHS Foundation Trust and NIHR Health Protection Research Unit (HPRU) in Behavioural Science and Evaluation at University of Bristol in partnership with Public Health England (PHE). Award number P315. The views expressed are those of the authors and not necessarily those of the NIHR, the Department of Health and Social Care or PHE.

**Competing interests** None declared.

**Patient consent for publication** Not applicable.

**Ethics approval** South West Frenchay Research Ethics Committee granted approval, reference 18/SW/0090.

**Provenance and peer review** Not commissioned; externally peer reviewed.

**Data availability statement** Data are available upon reasonable request. Data are available on request from corresponding author.

**ORCID iDs**
Ava Lorenc http://orcid.org/0000-0003-1132-4601
Emer Brangan http://orcid.org/0000-0002-1288-0960
Joanna M Kesten http://orcid.org/0000-0002-3674-6045

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
