## [Reviewer comments · BMJ Open]

ARTICLE DETAILS

TITLE (PROVISIONAL)	What can be learnt from a qualitative evaluation of implementing a rapid sexual health testing, diagnosis and treatment service?
AUTHORS	Lorenc, Ava; Brangan, Emer; Kesten, Joanna; Horner, Paddy; Clarke, Michael; Crofts, Megan; Steer, Jonathan; Turner, Jonathan; Muir, Peter; Horwood, Jeremy

VERSION 1 – REVIEW

REVIEWER	Bin Ibrahim , Muhamad Alif James Cook University Australia - Singapore Campus
REVIEW RETURNED	26-Apr-2021

GENERAL COMMENTS	Dear BMJ Open Editorial Office, Re: Review of manuscript number BMJOPEN-2021-050109 Thank you for the opportunity to review this manuscript! This paper is a qualitative evaluation study aimed at investigating the facilitators and barriers that staff and patients faced with a new sexual health service provided in a sexual health centre in Bristol, UK. The paper was appropriately grounded in the Normalisation Process Theory. It utilised an ethnographic approach (observations and participant interviews) to uncover the benefits of and challenges to the implementation of this service over sixteen months. As a qualitative researcher, I find that the undertaking of research in this paper commendable. The results offered important insights into the successful implementation of the first UK rapid NAAT testing for chlamydia and gonorrhoea. It also provided insights that could be used to implement similar services in other settings and inform other programs/process evaluations studies in the future. Below, I offer some feedback and suggestions that I hope will aid the authors in improving their paper. 1) Methods – Design • There are many designs and methods used for program and process evaluation research. Can the authors clarify the reasons for choosing ethnographic methods for this study to help the reader understand the method of choice?• The authors undertook observations and interviews at four time points over sixteen months. Could the authors clarify how the team decided on the number of time points and the length of follow-ups (0 months, six months, 14 months and 16 months)? 2) Methods – Participants • How was the number of required observations determined or planned in the first six months of service implementation? Were these based on pragmatic considerations or other factors?
---

	 • Who conducted the observations? Were these undertaken by the researchers who interviewed participants? • How were interview sample sizes for patient and staff participants determined? Were these determined a priori or determined as data collection and analysis proceeded? • I understand that Malterud et al. (2016) reference was cited after “Staff were emailed by the researcher about the study” on page 5. Can I clarify if this is an error and should have been mentioned after the following statement in that same paragraph? • The authors mentioned a maximum variation purposive sampling strategy but did not link this to any particular reference or citation. Perhaps this could be linked to the paper by Higginbottom et al. (2013) or Palinkas et al. (2015) - Purposeful sampling for qualitative data collection and analysis in mixed method implementation research. • While I commend the authors for using other strategies in determining their sample size besides saturation (the paper Higginbottom et al. (2013) had mentioned the use of data saturation as a strategy), there was no clarification on how information power was assessed. Malterud and colleagues (2016) explained five dimensions in their article to help determine the information power of one’s sample size: study aim, sample specificity, use of established theory, quality of dialogue, and analysis strategy. Could the authors explain how these items and dimensions were used to inform the sample size for the study? 3) Methods - Data Collection  • Were constructs from the Normalisation Process Theory also used in the construction of topics within the semi-structured interview guide? • The authors mentioned how staff were interviewed in four batches at each timepoint while patients were interviewed throughout the evaluation period. Can I clarify what this means – were patients only interviewed once throughout the 16 months while staff were interviewed at the different time points? Were the same participants interviewed again at the other time points? • Ethnographic work typically is accompanied by a reflexivity/positionality paragraph made by the authors. This importance was also mentioned in the paper by Higginbottom et al. (2013), which the authors have cited. Such statements or paragraphs would also help in strengthening the rigour and trustworthiness of their ethnographic approach to this study. • While the authors provided the use of the COREQ checklist at the end of the paper, can I clarify if there were other ways the authors have used to establish the rigour and trustworthiness of their findings? Perhaps this may tie in with my clarification on the authors' approach to thematic analysis under data analysis. 4) Methods – Data Analysis  • The authors mentioned, “ongoing and iterative analysis informed further data collection and service development”. Can the authors clarify how the ongoing analysis informed the data collection? Was the interview guide changed after each analysis? Were the services provided improved throughout the study period? How did this impact the data collection and analysis? • The authors mentioned “Thematic, inductive analysis” was conducted. However, the authors elaborated later that the constructs from the Normalisation Process Theory were used to develop the themes further. Thus, there seems to be a mixture of
--	---

	inductive and deductive processes going on here rather than merely inductive?  • Thematic analysis is also an umbrella of approaches with different procedures and underlying philosophical assumptions (please see Braun and Clarke (2020) – “One size fits all? What counts as quality practice in (reflexive) thematic analysis?). Can the authors clarify which approach in thematic analysis was used? The authors subsequently mentioned how a subset of transcripts was double coded and discussed for discrepancies or consensus – this seemed like a coding reliability approach (e.g. Boyatzis, 1998 or Guest et al., 2012). But I will leave this to the authors to clarify as they are more familiar with the analytical strategy used for their study. 5) Results  • The authors merely mentioned their participants' demographics in a summary paragraph at the beginning of the results section. A table containing the demographic information will help readers assess the maximum variation purposive sampling strategy that the authors utilised as a part of their study. • The authors mentioned the index of multiple deprivation scores in the “Participants and hours of observations section”. However, they made no mention of this index in the other parts of the paper and why this index was used or relevant in this study. 6) Discussion  • Were there any strengths and limitations regarding the methodologies that the authors have used in their study? For example, the triangulation of findings (a) between observations and interviews, and b) between the researchers in the team? These considerations would be beneficial for readers, which may help in future designs of other process and program evaluations. Thank you again for this opportunity to review this insightful study! I hope my comments and suggestions will help the authors to strengthen their paper further.
--	--

REVIEWER	Vujcich, Daniel Curtin University, School of Public Health
REVIEW RETURNED	27-Apr-2021

GENERAL COMMENTS	General comments  • This manuscript presents important research and the authors ought to be commended. However, the manuscript's current structure could be improved to enhance both the reader's experience and the impact of this research. Currently, the characteristics of the intervention are predominantly described in a table contained in the Results section. The paper would be clearer if: (a) the changes to the service delivery model were described at the outset, and; (b) the results section was limited to findings concerning the reception to, and consequences of, those changes. Introduction  • The introduction states: “much of the evidence is from modelling and hypothetical views of clinicians and/or patients^{11-13 26-29 31}, with little real-life implementation evaluation, and rarely considering the complexity of patient visits including both asymptomatic and symptomatic patients with multiple needs e.g. female contraception. There is an urgent need to evaluate staff and patient preferences, and clinical benefits and cost effectiveness in
--

practice". However, there are other relevant studies of STI POC testing that ought to at least be described briefly – e.g. see publications around the TTANGO2 study in Australian Indigenous populations: <https://www.ttango.com.au/publications>

- I think more description of the intervention is required (i.e. what did the shift to rapid testing involve for this service and how is it different to the way the service operated pre-intervention?). Currently this information is first mentioned in Table 2 but I think the flow of the paper would be assisted if these changes were introduced earlier (they are not a finding per se, but rather a description of the intervention).

Methods

- The methods should include information to explain why the chosen design was considered more appropriate than alternative approaches. What are the unique strengths of this approach?
- Please offer a brief explanation of NPT at the outset. The authors do include some information about NPT later in the methods (e.g. page 6) but even this could be explained in more detail, particularly in light of the heavy reliance on NPT concepts in the results section.
- More information about recruitment and data collection (specifically, observation) processes is needed to enhance replicability
- There is a need to include some information on how observation data were analysed (the description is currently limited to interviews)

Results

- Some information is needed to help assess whether those who agreed to participate had similar characteristics to those who refused.
- The findings currently read in quite a disjointed way. Some effort needs to be made to improve the flow between concepts. Alternatively, other ways of structuring the findings could be explored (e.g. perhaps readers would find it easier if the findings were presented in more of a narrative order: (1) pre-intervention preparation, communication and training; (2) testing patients under the new system; (3) communication of results under the new system; (4) treatment under the new system; (5) implications for other aspects of primary health – e.g. discussion of other health issues like |FGM, alcohol use etc; (5) general barriers, facilitators and recommendations for service improvements.
- It would be useful if some effort was made to integrate illustrative quotations into the text, rather than simply making it the reader's obligation to consult the tables which present long, unedited quotes.

Discussion

- The discussion would benefit from some engagement with organisational change literature more generally and more in-depth engagement with similar literature on the acceptability of rapid testing in other contexts.

Other

- Some content is expressed in a manner that is less formal than is normally expected in scholarly publications (e.g. reference to "admin" on page 6, "teething glitches" on page 10)
- The results section include some unusual uses of bolded text

VERSION 1 – AUTHOR RESPONSE

Comment	Response	Location
Reviewer 1		
1) Methods – Design There are many designs and methods used for program and process evaluation research. Can the authors clarify the reasons for choosing ethnographic methods for this study to help the reader understand the method of choice?	Thank you, we have clarified in the methods section that the ethnographic approach provides insights into peoples’ views and actions via the contexts and locations they inhabited and supported real-time feedback to refine and improve the service: The evaluation was ethnographic, theory-based (informed by Normalisation Process Theory (NPT) a sociological theory that has been widely promoted as a means to understand implementation, embedding and integration of innovation in healthcare settings¹) and used observations and interviews². This approach provided insights into peoples’ views and actions via the contexts and locations they inhabited³ and supported real-time feedback to refine and improve the service.	Para 1, Page 5
The authors undertook observations and interviews at four time points over sixteen months. Could the authors clarify how the team decided on the number of time points and the length of follow-ups (0 months, six months, 14 months and 16 months)?	We have clarified in the methods section that the timepoints were selected pragmatically rather than a-priori to capture the early and later phases of service implementation and the need to continue data collection until sufficient data had been obtained following the principles of information power.	Para 1 and 4, Page 5
2) Methods – Participants How was the number of required observations determined or planned in the first six months of service	Thank you for highlighting this, we agree more detail is needed to understand our data collection decisions. We have now included an overarching statement around the	Para 4, Page 5

implementation? Were these based on pragmatic considerations or other factors?	decision to end data collection and have clarified that the observations ‘capture the initial implementation challenges’: Data collection continued until sufficient data to meet the study objectives had been collected with continuous, pragmatic assessment of information within our sample ⁴.	
Who conducted the observations? Were these undertaken by the researchers who interviewed participants?	EB and JK conducted the observations. This information has been added.	Para 5, page 5
How were interview sample sizes for patient and staff participants determined? Were these determined a priori or determined as data collection and analysis proceeded?	We used the concept of information power to determine the sample size for staff and patient interviews and observations. We have now clarified this: Following the concept of information power, data collection continued until sufficient data to meet the study objectives had been collected with continuous, pragmatic assessment of information within our sample ⁴. Issues informing information power include the study aim (i.e. broader aims require a larger sample), the sample (i.e. a smaller sample is needed if participants have rich experiences relevant to the research), use of theory (studies supported by theory require smaller sample sizes), depth and quality of the data (i.e. smaller samples are needed with focused and clear data) and the analysis type (larger samples are needed for exploratory analysis) ⁴.	Para 4, Page 5
I understand that Malterud et al. (2016) reference was cited after “Staff were emailed by the researcher about the study” on page 5. Can I clarify if this is an error and should have been mentioned	Thank you for spotting this error. We have moved this sentence to the ‘Data collection’ section and further clarified the use of the Information Power concept (as above).	Para 4, Page 5

after the following statement in that same paragraph?		
The authors mentioned a maximum variation purposive sampling strategy but did not link this to any particular reference or citation. Perhaps this could be linked to the paper by Higginbottom et al. (2013) or Palinkas et al. (2015) - Purposeful sampling for qualitative data collection and analysis in mixed method implementation research.	Thank you for spotting this, we have referenced Higginbottom et al.	Para 3, Page 5
While I commend the authors for using other strategies in determining their sample size besides saturation (the paper Higginbottom et al. (2013) had mentioned the use of data saturation as a strategy), there was no clarification on how information power was assessed. Malterud and colleagues (2016) explained five dimensions in their article to help determine the information power of one's sample size: study aim, sample specificity, use of established theory, quality of dialogue, and analysis strategy. Could the authors explain how these items and dimensions were used to inform the sample size for the study?	We now include the following description of the dimensions of Information Power: Following the concept of information power, data collection continued until sufficient data to meet the study objectives had been collected with continuous, pragmatic assessment of information within our sample ⁴. Issues informing information power include the study aim (i.e. broader aims require a larger sample), the sample (i.e. a smaller sample is needed if participants have rich experiences relevant to the research), use of theory (studies supported by theory require smaller sample sizes), depth and quality of the data (i.e. smaller samples are needed with focused and clear data) and the analysis type (larger samples are needed for exploratory analysis) ⁴.	Para 4, Page 5
3) Methods - Data Collection Were constructs from the Normalisation Process Theory also used in the construction of topics within the semi-structured interview guide?	Yes, the interview topic guides were informed by NPT. We now state: Interview topic guides informed by NPT explored: views and experiences of the service; impact on workload and clinical practice; information and support needs, sustainability and future implementation of the service.	Para 6, page 5

The authors mentioned how staff were interviewed in four batches at each timepoint while patients were interviewed throughout the evaluation period. Can I clarify what this means – were patients only interviewed once throughout the 16 months while staff were interviewed at the different time points? Were the same participants interviewed again at the other time points? –	Patients were interviewed once throughout the study. One nurse who was interviewed twice, all other staff interviews occurred at one timepoint. To clarify, we have moved the sentence describing the staff interview batches to follow on from the equivalent sentence for the patient interviews and clarified that patient interviews took place in T1. One staff member was interviewed twice – this has been added: Patients were invited to take part, via a clinic survey about PrEP (pre-exposure prophylaxis for HIV) ⁵ and physically attending the clinic at T1, T2 and T3. Cross sectional staff interviews were conducted at four timepoints T1, T2, T3, T4. One staff member was interviewed twice.	Para 3, page 5
Ethnographic work typically is accompanied by a reflexivity/positionality paragraph made by the authors. This importance was also mentioned in the paper by Higginbottom et al. (2013), which the authors have cited. Such statements or paragraphs would also help in strengthening the rigour and trustworthiness of their ethnographic approach to this study.	The following reflexivity statement has been added to the discussion, under strengths and limitations: EB and JK came to the observations as experienced researchers and with good knowledge of the plans for the service changes and reasons for them. The researchers were surprised at how quickly it was possible to provide information and feedback to the implementation team which they clearly valued highly and rapidly implemented changes based on it. The researchers could move freely between different physical areas of the clinic and stages of the process in a way which clinic staff were not free to do, which provided early insights. Due to the study design and relationships, these insights could be discussed promptly with relevant staff - and so sense checked, and action taken in response if appropriate (changes to clinic processes; further data collection etc.). The rapid, supportive, evidence-	Para 4, page 12

	based feedback which the researchers could provide seemed to quickly build the confidence of the key implementation staff in the research process. The researchers appeared to be quickly accepted as trusted team members, with the capacity to help with the work at hand (rather than creating 'research burden').	
While the authors provided the use of the COREQ checklist at the end of the paper, can I clarify if there were other ways the authors have used to establish the rigour and trustworthiness of their findings? Perhaps this may tie in with my clarification on the authors' approach to thematic analysis under data analysis.	We include the following account relating to efforts to establish rigour and trustworthiness in the analysis section of the methods: Ongoing and iterative analysis informed further data collection through changes to the topic guide and service development, evaluation, adaptation, refinement and integration. A subset of transcripts and observations were independently double-coded by EB/JH and discrepancies discussed to contribute to the generation and refinement of codes to maximize rigour. Themes were discussed by the multi-disciplinary research team to ensure credibility and confirmability.	Para 2, Page 6
4) Methods – Data Analysis The authors mentioned, “ongoing and iterative analysis informed further data collection and service development”. Can the authors clarify how the ongoing analysis informed the data collection? Was the interview guide changed after each analysis? Were the services provided improved throughout the study period? How did this impact the data collection and analysis?	We have clarified that the ongoing analysis informed data collection through changes to the topic guides (see above) and have clarified that the analysis informed service delivery: Ongoing and iterative analysis informed further data collection through changes to the topic guide and feedback to healthcare staff to aid the	Para 2, Page 6

	adaptation and refinement of the rapid service.	
The authors mentioned “Thematic, inductive analysis” was conducted. However, the authors elaborated later that the constructs from the Normalisation Process Theory were used to develop the themes further. Thus, there seems to be a mixture of inductive and deductive processes going on here rather than merely inductive?	We have clarified that the use of the NPT in the analysis was deductive: The four NPT constructs ¹ were used to further develop themes deductively.	Para 2, page 6
Thematic analysis is also an umbrella of approaches with different procedures and underlying philosophical assumptions (please see Braun and Clarke (2020) – “One size fits all? What counts as quality practice in (reflexive) thematic analysis?). Can the authors clarify which approach in thematic analysis was used? The authors subsequently mentioned how a subset of transcripts was double coded and discussed for discrepancies or consensus – this seemed like a coding reliability approach (e.g. Boyatzis, 1998 or Guest et al., 2012). But I will leave this to the authors to clarify as they are more familiar with the analytical strategy used for their study.	Our analytic approach most closely aligns with what Braun and Clarke refer to as ‘codebook’ TA in their 2020 paper:: ‘Codebook’ thematic, inductive analysis by EB/AL identified and analysed patterns and themes salient to participants and observations ⁶ .	Para 2, page 6
5) Results The authors merely mentioned their participants' demographics in a summary paragraph at the beginning of the results section. A table containing the demographic information will help readers assess the maximum variation purposive sampling strategy that the authors utilised as a part of their study.	To protect participant anonymity, we decided not to provide individual level demographic information about staff and patients. This was particularly a concern for staff who are part of a small team which risks identifying who they are. We feel the demographic information provided at the start of the results section is sufficiently detailed to demonstrate the diversity of the sample: Patients ranged in age from 19 to 57, average 34 years, index of multiple deprivation scores ranged from 2 to 10, average 5.4, and most identified as MSM. Two had positive STI test results.	Para 5, page 6

The authors mentioned the index of multiple deprivation scores in the “Participants and hours of observations section”. However, they made no mention of this index in the other parts of the paper and why this index was used or relevant in this study.	We present the IMD scores to illustrate the diversity of our interview sample, in the same way that we report age and STI test result.	
6) Discussion  Were there any strengths and limitations regarding the methodologies that the authors have used in their study? For example, the triangulation of findings (a) between observations and interviews, and b) between the researchers in the team? These considerations would be beneficial for readers, which may help in future designs of other process and program evaluations. 	Thank you for highlighting this. We have added a heading to the strengths and limitations paragraph so it stands out and have added: Project strengths include: integration of findings from multiple qualitative methods generating rich insights, a multidisciplinary team including clinical academics; a strong trusting relationship between research team and clinical staff due to existing relationships and research team flexibility and responsiveness; regular feedback from researchers to clinicians using a ‘trial, assess, adapt’ strategy.	Para 4, page 12
Thank you again for this opportunity to review this insightful study! I hope my comments and suggestions will help the authors to strengthen their paper further.	Thank you very much for your time and effort in reviewing the manuscript. The co-authors and I appreciate the comments received.	
Reviewer: 2		
General comments  This manuscript presents important research and the authors ought to be commended. However, the manuscript's current structure could be improved to enhance both the reader's experience and the impact of this research. Currently, the characteristics of the intervention are predominantly described in a table contained in the Results section. The paper would be clearer if: (a) the changes to the service delivery model were described at the outset, and; (b) the 	We thank the reviewer for their useful suggestions. We have included a flow diagram (Figure 1) in the introduction section to make the service delivery model before and after the implementation of the rapid testing system clear. The results section charts the implementation of the new service and includes details of the changes to the	Para 4, page 4

results section was limited to findings concerning the reception to, and consequences of, those changes.	service as well as the response to and consequences of the changes. Changes were made iteratively in response to our rapid ethnography and therefore we are classing these as results.	
Introduction  The introduction states: “much of the evidence is from modelling and hypothetical views of clinicians and/or patients^{11-13 26-29 31}, with little real-life implementation evaluation, and rarely considering the complexity of patient visits including both asymptomatic and symptomatic patients with multiple needs e.g. female contraception. There is an urgent need to evaluate staff and patient preferences, and clinical benefits and cost effectiveness in practice”. However, there are other relevant studies of STI POC testing that ought to at least be described briefly – e.g. see publications around the TTANGO2 study in Australian Indigenous populations: https://www.ttango.com.au/publications 	Thank you for highlighting the TTANGO2 study in Australian Indigenous population. We now cite the following qualitative acceptability study: “I Do Feel Like a Scientist at Times”: A Qualitative Study of the Acceptability of Molecular Point-Of-Care Testing for Chlamydia and Gonorrhoea to Primary Care Professionals in a Remote High STI Burden Setting Natoli L, Guy RJ, Shephard M, Causer L, Badman SG, et al. (2015) “I Do Feel Like a Scientist at Times”: A Qualitative Study of the Acceptability of Molecular Point-Of-Care Testing for Chlamydia and Gonorrhoea to Primary Care Professionals in a Remote High STI Burden Setting. PLOS ONE 10(12): e0145993. https://doi.org/10.1371/journal.pone.0145993	Introduction
I think more description of the intervention is required (i.e. what did the shift to rapid testing involve for this service and how is it different to the way the service operated pre-intervention?). Currently this information is first mentioned in Table 2 but I think the flow of the paper would be assisted if these changes were introduced earlier (they are not a finding per se, but rather a description of the intervention). -	As above, we now include a flow diagram (Figure 1) in the introduction section to make the service delivery model before and after the implementation of the rapid testing system clear.	Para 4, page 4
Methods  The methods should include information to explain why the chosen design was considered more appropriate 	Thank you. In response to reviewer 1 we have clarified the strength of the approach taken: The evaluation was ethnographic, theory-based (informed by	Para 5, page 4

than alternative approaches. What are the unique strengths of this approach?	Normalisation Process Theory (NPT) a sociological theory that has been widely promoted as a means to understand implementation, embedding and integration of innovation in healthcare settings ¹⁾ and used observations and interviews². This approach provided insights into peoples' views and actions via the contexts and locations they inhabited³ and supported real-time feedback to refine and improve the service. Unfortunately, the word limit does not allow for a fuller description of the alternative approaches.	
Please offer a brief explanation of NPT at the outset. The authors do include some information about NPT later in the methods (e.g. page 6) but even this could be explained in more detail, particularly in light of the heavy reliance on NPT concepts in the results section.	As above, the following has been added to explain NPT at the outset: The evaluation was ethnographic, theory-based (informed by Normalisation Process Theory (NPT) a sociological theory that has been widely promoted as a means to understand implementation, embedding and integration of innovation in healthcare settings ¹⁾ and used observations and interviews².	Para 5, page 4
More information about recruitment and data collection (specifically, observation) processes is needed to enhance replicability.	We now include more detail on the observations: ...Observations were conducted by EB and JK at varying times/days, in reception, laboratory and waiting areas. Non-participant observations focussed on day-to-day operations, how clinic staff integrated the new service and any factors which promoted or inhibited successful incorporation ³. Written accounts based on brief field notes taken at the time included observations, conversations with staff, and reflection on what has been observed⁷. Observations recorded activities, events, their time and location and	Para 5, page 5

	described interactions, communication patterns, workflows and tasks in the Unity clinic environment.	
There is a need to include some information on how observation data were analysed (the description is currently limited to interviews)	We have clarified that the observation data were analysed alongside the interview data: ‘Codebook’ thematic, inductive analysis by EB/AL identified and analysed patterns and themes salient to participants and observations ⁶. Initial noting of ideas was followed by line-by-line examination and inductive coding. A subset of transcripts and observations were independently double-coded by EB/JH and discrepancies discussed.	Para 2, page 6
Results Some information is needed to help assess whether those who agreed to participate had similar characteristics to those who refused.	The characteristics of our sample are presented and demonstrate its diversity. Unfortunately, we do not know the details of those who did not agree to participate.	
The findings currently read in quite a disjointed way. Some effort needs to be made to improve the flow between concepts. Alternatively, other ways of structuring the findings could be explored (e.g. perhaps readers would find it easier if the findings were presented in more of a narrative order: (1) pre-intervention preparation, communication and training; (2) testing patients under the new system; (3) communication of results under the new system; (4) treatment under the new system; (5) implications for other aspects of primary health – e.g. discussion of other health issues like FGM, alcohol use etc; (5) general barriers, facilitators and recommendations for service improvements.	We thank the reviewer for their ideas for structuring the results. As the other reviewer did not have an issue with how the results were structured, we have decided not to change the structure of the results. The evaluation took a theory-informed approach drawing on normalisation process theory (NPT) which informed the analysis, and the findings are structured in accordance with NPT. The way we have structured the results is the accepted way of presenting findings using NPT, here are some other published examples that use the same structure:	

	 • https://implementationscience.biomedcentral.com/articles/10.1186/s13012-015-0230-4 • https://academic.oup.com/fampra/article/33/6/704/2503159#49040543 • https://implementationscience.biomedcentral.com/articles/10.1186/1748-5908-7-106 • https://bmcfampract.biomedcentral.com/articles/10.1186/s12875-016-0453-8 • https://bmcpneumology.biomedcentral.com/articles/10.1186/s12888-016-0761-5#Sec8 • https://www.nature.com/articles/bjc201546#Sec7 • https://pubmed.ncbi.nlm.nih.gov/32094220/ 	
 • It would be useful if some effort was made to integrate illustrative quotations into the text, rather than simply making it the reader's obligation to consult the tables which present long, unedited quotes. 	Unfortunately, due to the word limit of the journal, we are unable to integrate the quotes within the main text.	
Discussion  • The discussion would benefit from some engagement with organisational change literature more generally and more in-depth engagement with similar literature on the acceptability of rapid testing in other contexts. 	We have expanded our engagement with the rapid testing organisational change literature in relation to sexual health in the discussion and have included 2021 literature that has been published since we submitted our paper. As we are already at the word limit for the paper so do not have the space for in-depth engagement in literature from other contexts. Evaluating the real-life implementation of a novel rapid results service confirms previous hypothetical/simulated studies where patients were happy with the service and willing to wait for results before treatment⁸⁻¹¹. Whereas previous research has found that the patients found the hypothetical scenario of waiting up to 40 minutes for test results acceptable^{11 12}, our findings demonstrate that patients were happy to wait up to 48 hours for treatment based on results. Willingness to wait	Para 2, page 11

	has been found to be dependent on self-assessed infection risk and anxiety about their infection status ¹¹. Our findings demonstrate that the rapid service can lead to less patient anxiety due to shorter time waiting for results and therefore should target patients concerned they are infected. Although asymptomatic patients are encouraged to use on-line postal services, some patients may wish to attend in-person clinics^{13 14}. The benefits of treating with results and improving antimicrobial stewardship previously anticipated^{15 16} are highly valued by staff and patients in our evaluation. Modelling studies have demonstrated that rapid testing can enable faster treatment, reduces infectious periods, and leads to fewer transmissions, partner attendances and clinic costs^{17 18}. Rapid diagnostics and treatment can increase the proportion of individuals receiving timely treatment and decrease community prevalence of STIs^{19 20} and recently has been seen as a key factor contributing to the reducing new HIV infections in London and ensuring those with HIV receive fast and optimal care²¹. Our findings also confirm reductions in patient anxiety^{15 16 22} and improved testing uptake²³⁻²⁵ are likely, as well as freeing up clinician time, greater clinician confidence, and efficiencies allowing capacity to be utilised elsewhere¹⁵.	
Other  Some content is expressed in a manner that is less formal than is normally expected in scholarly publications (e.g. reference to “admin” on page 6, “teething glitches” on page 10) 	We have changed the language throughout the paper, for example: Teething glitches has been replaced by ‘initial challenges.’ Admin has been replaced by ‘administration’ or ‘administrative’ as appropriate.	

The results section include some unusual uses of bolded text	Thank you. We have removed the bold formatting.	
--	---	--

1. Murray E, Treweek S, Pope C, et al. Normalisation process theory: a framework for developing, evaluating and implementing complex interventions. *BMC medicine* 2010;8:63. doi: 10.1186/1741-7015-8-63 [published Online First: 2010/10/22]
2. Higginbottom GM, Pillay JJ, Boadu NY. Guidance on Performing Focused Ethnographies with an Emphasis on Healthcare Research *The Qualitative Report* 2013;18(9):1-6.
3. Reeves S, Kuper A, Hodges BD. Qualitative research methodologies: ethnography. *BMJ* 2008;337 doi: 10.1136/bmj.a1020
4. Malterud K, Siersma VD, Guassora AD. Sample Size in Qualitative Interview Studies: Guided by Information Power. *Qualitative Health Research* 2016;26(13):1753-60. doi: 10.1177/1049732315617444
5. MacGregor L, Speare N, Nicholls J, et al. The role of the evolving trends in sexual behaviours and clinic attendance patterns on increasing diagnoses of STIs: a perspective from Bristol, UK. *Sexually Transmitted Infections* 2020; Accepted for Publication
6. Braun V, Clarke V. One size fits all? What counts as quality practice in (reflexive) thematic analysis? *Qualitative Research in Psychology* 2020:1-25. doi: 10.1080/14780887.2020.1769238
7. Emerson R, Fretz R, Shaw L. *Writing ethnographic fieldnotes*. Chicago and London: University of Chicago Press 1995.
8. Llewellyn CD, Sakal C, Lagarde M, et al. Testing for sexually transmitted infections among students: a discrete choice experiment of service preferences. *BMJ Open* 2013;3(10) doi: 10.1136/bmjopen-2013-003240
9. Chislett L, Clarke J. Which elements of a novel self-directed rapid asymptomatic sexually transmitted infection screening service are most important to users? . *Sexually Transmitted Infections*, 2015:A1–A258.
10. Mahilum-Tapay L, Laitila V, Wawrzyniak JJ, et al. New point of care Chlamydia Rapid Test--bridging the gap between diagnosis and treatment: performance evaluation study. *BMJ (Clinical research ed)* 2007;335(7631):1190-94. doi: 10.1136/bmj.39402.463854.AE [published Online First: 11/30]
11. Fuller SS, Pacho A, Broad CE, et al. "It's not a time spent issue, it's a 'what have you spent your time doing?' issue. . ." A qualitative study of UK patient opinions and expectations for implementation of Point of Care Tests for sexually transmitted infections and antimicrobial resistance. *PLoS one* 2019;14(4):16. doi: 10.1371/journal.pone.0215380
12. Widdice L, Hsieh YH, Silver B, et al. Performance of the Atlas Genetics Rapid Test for Chlamydia trachomatis and women's attitudes toward point-of-care testing. *Sexually Transmitted Diseases* 2018;45(11):5. doi: 10.1097/OLQ.0000000000000865
13. Whitlock GG, Gibbons DC, Longford N, et al. Rapid testing and treatment for sexually transmitted infections improve patient care and yield public health benefits. *International journal of STD & AIDS* 2018;29(5):14. doi: 10.1177/0956462417736431 [published Online First: 2017/10/24]
14. Turner KME, Zienkiewicz AK, Syred J, et al. Web-Based Activity Within a Sexual Health Economy: Observational Study. *J Med Internet Res* 2018;20(3):e74. doi: 10.2196/jmir.8101
15. Adams EJ, Ehrlich A, Turner KME, et al. Mapping patient pathways and estimating resource use for point of care versus standard testing and treatment of chlamydia and gonorrhoea in genitourinary medicine clinics in the UK. *BMJ Open* 2014;4(7) doi: 10.1136/bmjopen-2014-005322
16. Turner KME, Round J, Horner P, et al. An early evaluation of clinical and economic costs and benefits of implementing point of care NAAT tests for Chlamydia trachomatis and Neisseria gonorrhoea in genitourinary medicine clinics in England. *Sexually Transmitted Infections* 2013 doi: 10.1136/sextrans-2013-051147
17. Mohiuddin S, Gardiner R, Crofts M, et al. Modelling patient flows and resource use within a sexual health clinic through discrete event simulation to inform service redesign. *BMJ Open* 2020;10(7):e037084. doi: 10.1136/bmjopen-2020-037084 [published Online First: 2020/07/10]

18. Whitlock GG, Gibbons DC, Longford N, et al. Rapid testing and treatment for sexually transmitted infections improve patient care and yield public health benefits. *International journal of STD & AIDS* 2018;29(5):474-82. doi: 10.1177/0956462417736431 [published Online First: 2017/10/24]
19. Keizur EM, Goldbeck C, Vavala G, et al. Safety and Effectiveness of Same-Day Chlamydia trachomatis and Neisseria gonorrhoeae Screening and Treatment Among Gay, Bisexual, Transgender, and Homeless Youth in Los Angeles, California, and New Orleans, Louisiana. *Sex Transm Dis* 2020;47(1):19-23. doi: 10.1097/olq.0000000000001088 [published Online First: 2019/11/07]
20. Natoli L, Guy RJ, Shephard M, et al. Public health implications of molecular point-of-care testing for chlamydia and gonorrhoea in remote primary care services in Australia: a qualitative study. *BMJ Open* 2015;5(4):e006922. doi: 10.1136/bmjopen-2014-006922 [published Online First: 2015/04/30]
21. Girometti N, Delpech V, McCormack S, et al. The success of HIV combination prevention: The Dean Street model. *HIV medicine* 2021 doi: 10.1111/hiv.13149 [published Online First: 2021/07/30]
22. Llewellyn C, Pollard A, Miners A, et al. Understanding patient choices for attending sexually transmitted infection testing services: a qualitative study. *Sex Transm Infect* 2012;88(7):504-9. doi: 10.1136/sextrans-2011-050344 [published Online First: 2012/05/26]
23. Natoli L, Guy RJ, Shephard M, et al. Public health implications of molecular point-of-care testing for chlamydia and gonorrhoea in remote primary care services in Australia: a qualitative study. *BMJ Open* 2015;5(4) doi: 10.1136/bmjopen-2014-006922
24. Rompalo AM, Hsieh Y-H, Hogan T, et al. Point-of-care tests for sexually transmissible infections: what do 'end users' want? *Sexual health* 2013;10(6):541-45. doi: 10.1071/SH13047
25. Horwood J, Ingle SM, Burton D, et al. Sexual health risks, service use, and views of rapid point-of-care testing among men who have sex with men attending saunas: a cross-sectional survey. *International journal of STD & AIDS* 2016;27(4):273-80. doi: 10.1177/0956462415580504 [published Online First: 2015/04/25]

VERSION 2 – REVIEW

REVIEWER	Bin Ibrahim , Muhamad Alif James Cook University Australia - Singapore Campus
REVIEW RETURNED	13-Aug-2021

GENERAL COMMENTS	Thank you for the opportunity to review the revised manuscript! I have looked through the comments and changes made by the author and co-authors. I would like to commend the team for the time and effort put into addressing the comments and suggestions from the two reviewers, especially working within the word limits that the journal has laid out. The team has addressed all of my previous comments and suggestions. I have no further suggestions for consideration. Thank you again for allowing me to review this insightful study!
--

REVIEWER	Vujcich, Daniel Curtin University, School of Public Health
REVIEW RETURNED	17-Aug-2021

GENERAL COMMENTS	Thank you for the opportunity to review the revised version of the manuscript. I commend the authors for the changes they have made which have resulted in a stronger manuscript. I continue to feel that this is an important study with both academic merit and practical significance.
--

Prior to recommending this manuscript be accepted for publication, I ask that the following be addressed:

(1) I remain of the view that more description of NPT is needed from the outset, given (a) its centrality to this manuscript and, (b) the fact that many readers of a BMJ journal will not necessarily be familiar with the theory (even though audiences of sociological journals may be). For examples of other articles in health/medical journals that provide more appropriate amounts of explanation, see:

<https://journals.plos.org/plosone/article?id=10.1371/journal.pone.0177026>

<https://link.springer.com/article/10.1186/1741-7015-8-63>

<https://www.sciencedirect.com/science/article/pii/S0020748913002010>

At the very least, I would recommend moving the four NPT constructs out of the analysis section and into the design section of the paper.

(2) I remain of my earlier view that “[t]he findings currently read in quite a disjointed way. Some effort needs to be made to improve the flow between concepts.” I understand and accept the authors’ reasoning for using NPT as the framework for the findings. However, some paragraphs in the results section continue to comprise of only one sentence; this creates a jarring experience for the reader and greater attention needs to be given to refining the flow of concepts and ideas.

Additionally, please note the manuscript still contains a number of typographical errors and grammatical issues, including:

- “Cross sectional interviewed were conducted”
- “25 observations were conducted, approximately 40 hours total, 25 staff interviews (24 participants), 26 patient interviews.”
- “Engagement was also limited by a lack of protected project time – implementation work was fitted around existing high workloads, rapid changes made timely feedback difficult, and burn out from staff pressures (funding cuts, understaffing, and high service demand).”
- “Rapid diagnostics and treatment can in increase”
- Some informal expressions remain, e.g. “lab”

In the authors’ reply to my previous feedback, I note that they cite word count constraints as a justification for not undertaking recommended revisions. I am empathetic to the reality that journal word counts are a constraint. However, before I am able to recommend this article for publication in BMJ Open, the authors either need to address the feedback within the journal’s word count or, alternatively, apply to the editor for permission to exceed the word count in light of the changes that have been requested.

I believe that the manuscript will be more useful and impactful if the changes I recommend are able to be adopted.

VERSION 2 – AUTHOR RESPONSE

Comment	Response
Reviewer 1	
Mr. Muhamad Alif Bin Ibrahim, James Cook University Australia - Singapore Campus Comments to the Author: Thank you for the opportunity to review the revised manuscript! I have looked through the comments and changes made by the author and co-authors. I would like to commend the team for the time and effort put into addressing the comments and suggestions from the two reviewers, especially working within the word limits that the journal has laid out. The team has addressed all of my previous comments and suggestions. I have no further suggestions for consideration. Thank you again for allowing me to review this insightful study!	Thank you for your positive feedback. We are pleased you enjoyed the paper.
Reviewer: 2 Dr. Daniel Vujcich, Curtin University Comments to the Author: Thank you for the opportunity to review the revised version of the manuscript. I commend the authors for the changes they have made which have resulted in a stronger manuscript. I continue to feel that this is an important study with both academic merit and practical significance.	Thank you for your positive feedback.
(1) I remain of the view that more description of NPT is needed from the outset, given (a) its centrality to this manuscript and, (b) the fact that many readers of a BMJ journal will not necessarily be familiar with the theory (even though audiences of sociological journals may be). For examples of other articles in health/medical journals that provide more appropriate amounts of explanation, see: https://journals.plos.org/plosone/article?id=10.1371/journal.pone.0177026 https://link.springer.com/article/10.1186/1741-7015-8-63 https://www.sciencedirect.com/science/article/pii/S0020748913002010	Thank you. We agree that having more detail on NPT is useful for unfamiliar readers. We expand the details on NPT and have also moved the four NPT constructs to the design section of the Methods on page 4 and have included the Murray reference (34). This is now in line with the reporting of the O'Reilly reference which employs NPT in a similar way.

At the very least, I would recommend moving the four NPT constructs out of the analysis section and into the design section of the paper.	
(2) I remain of my earlier view that “[t]he findings currently read in quite a disjointed way. Some effort needs to be made to improve the flow between concepts.” I understand and accept the authors’ reasoning for using NPT as the framework for the findings. However, some paragraphs in the results section continue to comprise of only one sentence; this creates a jarring experience for the reader and greater attention needs to be given to refining the flow of concepts and ideas.	Thank you for this feedback. We have revised the results section to improve the flow as recommended.
Additionally, please note the manuscript still contains a number of typographical errors and grammatical issues, including:  - “Cross sectional interviewed were conducted” - “25 observations were conducted, approximately 40 hours total, 25 staff interviews (24 participants), 26 patient interviews.” - “Engagement was also limited by a lack of protected project time – implementation work was fitted around existing high workloads, rapid changes made timely feedback difficult, and burn out from staff pressures (funding cuts, understaffing, and high service demand).” - “Rapid diagnostics and treatment can in increase” - Some informal expressions remain, e.g. “lab” 	Thank you for identifying these errors. We have thoroughly proof read the manuscript and hope we have detected all errors.

VERSION 3 – REVIEW

REVIEWER	Vujcich, Daniel Curtin University, School of Public Health
REVIEW RETURNED	24-Sep-2021
GENERAL COMMENTS	Thank you for your patience and commitment to accommodating my suggestions for improvement. I think the paper now reads very well and is an important contribution to the field. Congratulations.